# Facile accelerated specific therapeutic (FAST) platform develops antisense therapies to counter multidrug-resistant bacteria

Kristen A. Eller[1], Thomas R. Aunins[1], Colleen M. Courtney[1,2], Jocelyn K. Campos[1], Peter B. Otoupal[1], Keesha E. Erickson[1], Nancy E. Madinger[3] & Anushree Chatterjee[1,2,4,5 ✉]

Multidrug-resistant (MDR) bacteria pose a grave concern to global health, which is perpetuated by a lack of new treatments and countermeasure platforms to combat outbreaks or antibiotic resistance. To address this, we have developed a Facile Accelerated Specific Therapeutic (FAST) platform that can develop effective peptide nucleic acid (PNA) therapies against MDR bacteria within a week. Our FAST platform uses a bioinformatics toolbox to design sequence-specific PNAs targeting non-traditional pathways/genes of bacteria, then performs in-situ synthesis, validation, and efficacy testing of selected PNAs. As a proof of concept, these PNAs were tested against five MDR clinical isolates: carbapenem-resistant *Escherichia coli*, extended-spectrum beta-lactamase *Klebsiella pneumoniae*, New Delhi Metallo-beta-lactamase-1 carrying *Klebsiella pneumoniae*, and MDR *Salmonella enterica*. PNAs showed significant growth inhibition for 82% of treatments, with nearly 18% of treatments leading to greater than 97% decrease. Further, these PNAs are capable of potentiating antibiotic activity in the clinical isolates despite presence of cognate resistance genes. Finally, the FAST platform offers a novel delivery approach to overcome limited transport of PNAs into mammalian cells by repurposing the bacterial Type III secretion system in conjunction with a kill switch that is effective at eliminating 99.6% of an intracellular *Salmonella* infection in human epithelial cells.

[1] Chemical and Biological Engineering, University of Colorado Boulder, Boulder, CO 80303, USA. [2] Sachi Bioworks, Inc, Boulder, CO 80301, USA. [3] Division of Infectious Diseases, University of Colorado Denver, Aurora, CO 80045, USA. [4] Biomedical Engineering, University of Colorado Boulder, Boulder, CO 80303, USA. [5] Antimicrobial Regeneration Consortium, Boulder, CO 80301, USA. ✉email: chatterjee@colorado.edu

The development of antibiotic resistance is an emerging crisis in worldwide public health[1–4] and is outpacing the pipeline of new small molecule drugs[5–7]. The large majority of drugs in clinical development for priority pathogens—as defined by the World Health Organization[2]—do not introduce new classes or targets and are not pathogen-specific[5], thus presenting higher risk of rapid bacterial adaptation[1,8–12]. Due to rapid, pervasive bacterial adaption there is a need for countermeasure platforms that can generate antimicrobial solutions in a facile and accelerated manner. Antisense therapeutics offer the potential for gene sequence-specific therapies that can target a broader range of pathways in bacteria, are faster to design, and are more adaptive to resistance than conventional small molecule antibiotics[13,14]. Peptide nucleic acids (PNA) present an antisense strategy that offer advantages in stability[15,16], binding strength[17], and mismatch discrimination compared to similar technologies[18]. Prior research from our lab and others have demonstrated the utility of PNA as an antibiotic platform[19–24].

Despite these advantages, previous PNA work has relied on tedious design and screening processes to maximize stability and specificity of PNAs[20,25,26]. In addition, the inefficient transport of PNA across cell membranes, as well as the lack of a systematic effort to target non-traditional antibiotic pathways, has limited the development of PNA as useful antibiotics. We address these limitations through the introduction of Facile Accelerated Specific Therapeutic (FAST)[27], a semi-automated platform for the quick and efficient design, synthesis, testing, and delivery of PNA antisense antibiotics (Fig. 1). We are able to complete this full process in under 5 days. The FAST platform uses the PNA Finder toolbox to design PNA candidate sequences in less than 10 min, which are then synthesized, purified, and tested in parallel on a panel of multidrug-resistant (MDR) clinical isolates to validate the toolbox's predictions. These results can then be recycled to the PNA Finder toolbox to improve upon specificity predictions. Finally, promising PNAs are delivered to treat intracellular infections of bacteria.

## Results and discussion

**Bioinformatics toolbox designs PNA targeting five MDR Enterobacteriaceae clinical isolates.** The PNA Finder bioinformatics toolbox offers the functions Get Sequences and Find Off-Targets, which comprise of automated workflows that combine custom Python 3.7 scripts with the alignment and analysis programs Bowtie 2[28], SAMTools[29], and BEDTools[30]. Get Sequences is used to create a library of PNA candidate sequences that target the mRNA translation start codons of a user-defined set of genes. The function takes a list of gene IDs and searches a genome annotation file for matches, which can then be used to extract nucleotide sequences from a corresponding genome assembly. In addition, Get Sequences provides sequence warnings for PNA solubility and self-complementarity issues, as well as a protein interaction network analysis via the STRING database[31]. The PNA sequences provided by Get Sequences are used as inputs for the Find Off-Targets function, which searches for incidental inhibitory alignments by searching a non-target genome for highly similar sequences—1-bp mismatch or less—within a 20 nucleotide range of the start codon[26]. This function allows for the design of highly specific PNAs that target only the desired gene in the specific pathogens of interest, while avoiding broad inhibitory action against other pathogens, the human microbiome, and the human transcriptome.

In this study we applied the FAST platform[27] for the creation of PNA antibiotics against five MDR Enterobacteriaceae clinical isolates: two *Escherichia coli* isolates, two *Klebsiella pneumoniae* isolates, and one *Salmonella enterica* serovar Typhimurium isolate. To design the set of candidate sequences, the Get Sequences tool was used with a list of 296 essential and 4090 non-essential *E. coli* genes, as well as the genome assembly and annotations for reference strain *E. coli* MG1655 (Fig. 2a). The Find Off-Targets tool was used with this list to find off-targets within the *E. coli* genome, in order to avoid inhibition of unintended genes that could confound results. After filtering all

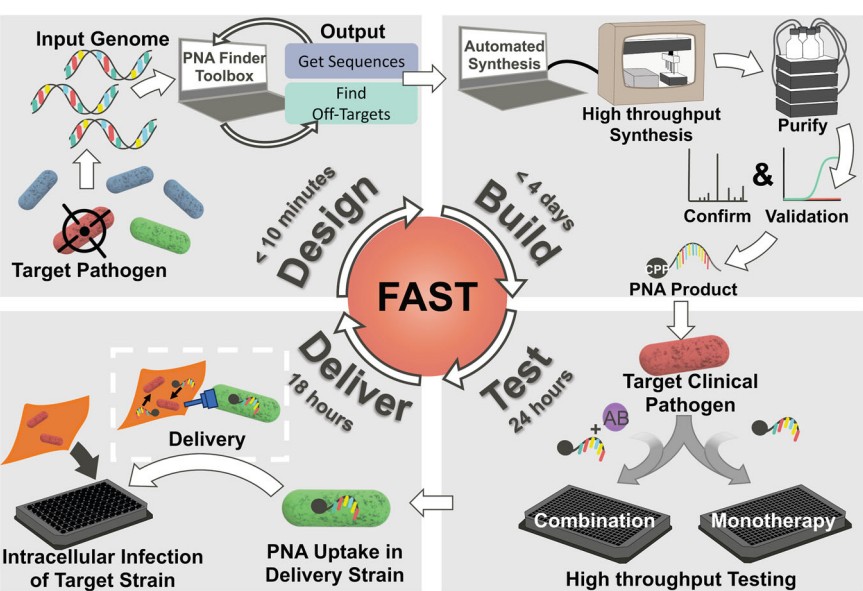

**Fig. 1 Facile Accelerated Specific Therapeutic (FAST) platform from design to delivery of PNA to treat MDR bacteria in less than 1 week.** The FAST pipeline is capable of producing effective therapies against MDR bacteria by combining design, synthesis, testing, and delivery modules. Design (top left corner): The corresponding reference genome of target pathogen(s) is inputted into the PNA Finder bioinformatics toolbox which generates a list of PNA sequences in less than 10 min. Build (top right corner): Promising PNA candidates are synthesized using high throughput, automated solid-phase synthesis chemistry. Purification of the PNA product is done using HPLC and validation is performed through LC–MS. The entire process is completed in less than 4 days. Test (bottom right corner): The purified and verified PNA is tested on clinical isolates within 1 day to evaluate therapeutic potential for monotherapy or combination (with antibiotic (AB)) treatments. Delivery (bottom left corner) Top PNA candidates are delivered to infected mammalian cells by repurposing the bacterial type III secretion system (T3SS) mediated delivery system to clear a secondary (red bacteria) intracellular infection.

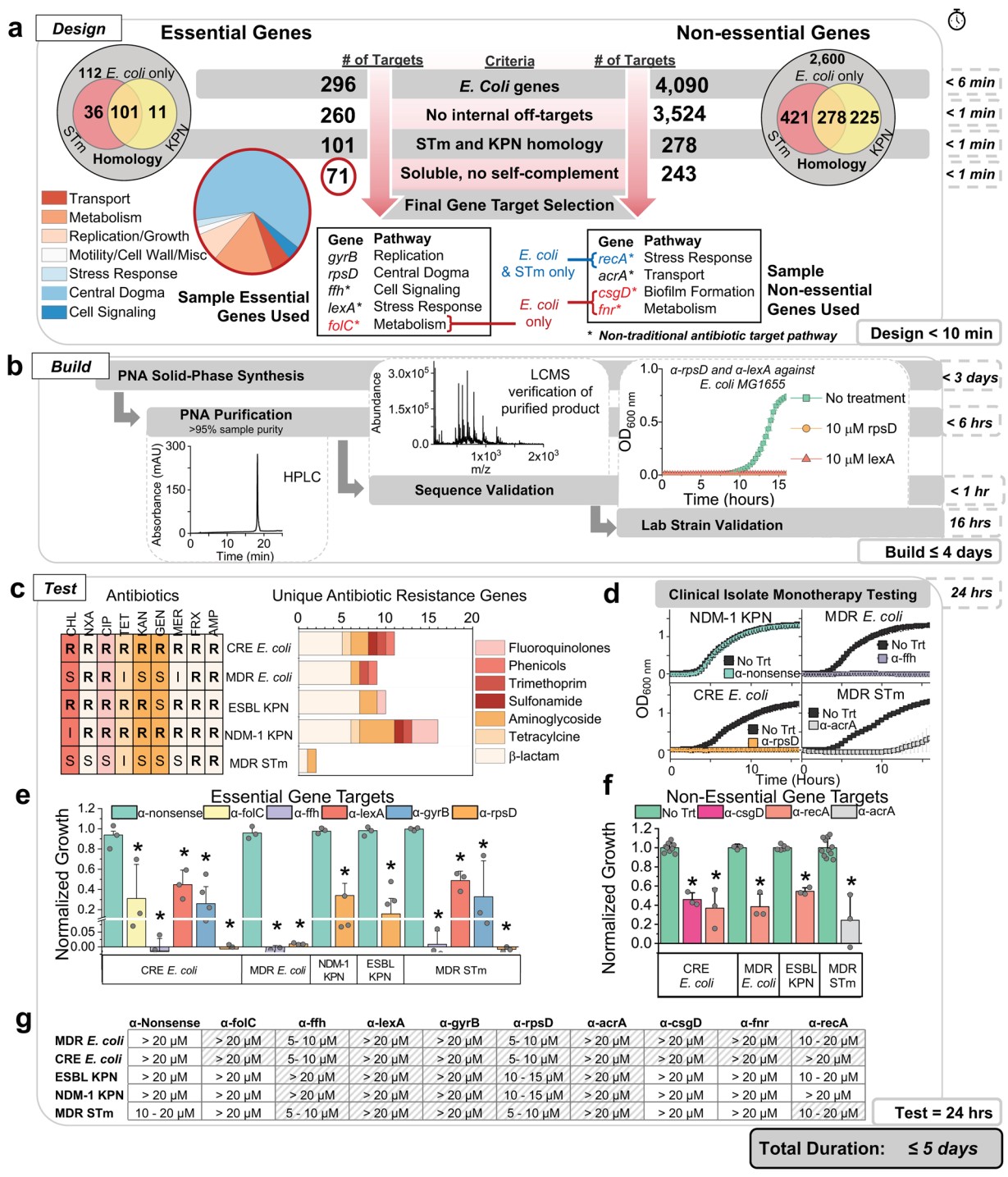

PNA with 0-bp mismatch off-targets we were left with a library of 260 PNA candidates targeting essential genes and 3524 candidates targeting non-essential genes within *E. coli*. The alignments of PNA candidates to these genomes were analyzed to determine which would be predicted to target *K. pneumoniae* and *S. Typhimurium* genes that are analogous to the original *E. coli* gene target. We ran this list of candidates through Find Off-Targets twice: first with *K. pneumoniae* (KPN) reference genome MGH 78578, then *Salmonella enterica* serovar Typhimurium (STm) reference genome SL1344. From this analysis we identified 101 essential and 278 non-essential PNA candidates that showed target sequence homology between the three reference genomes. Finally, we eliminated PNAs that were predicted by the Get

Sequences tool to have low solubility or exhibit self-complementarity, resulting in a list of 71 essential and 243 non-essential target gene candidates. The duration of this computational design pipeline was under 10 minutes.

For testing with the MDR isolates, we first selected two novel gene targets with functions related to those inhibited by conventional antibiotics: *gyrB*[32] (protein gyrase subunit B) and *rpsD*[33] (30 S ribosomal protein S4) (Fig. 2a). These genes contribute to pathways similar to those inhibited by fluoroquinolones and tetracycline/aminoglycosides respectively. We then chose three non-conventional antibiotic gene targets: *ffh*[34] (signal recognition particle protein, essential), *lexA*[35] (SOS response repressor protein, essential), and *acrA*[36] (transmembrane

**Fig. 2 Bioinformatics design of PNAs used to synthesis, purification, and monotherapy testing. a** Bioinformatic toolbox predicts PNAs that can target essential and non-essential genes in one or more Enterobacteriaceae including *E. coli*, *K. pneumoniae* (KPN), and *S. enterica* (STm). Here, PNAs targeting *E. coli* genome are identified and screened for internal off-targets. Candidates without off-targets are narrowed to those with homology among KPN and STm. Of the final 71 essential gene candidates and 243 non-essential gene candidates that met the thermodynamic requirements for experimental conditions, five and four PNAs targeting essential and non-essential gene respectively were randomly chosen for assessment. Most of the PNAs have homology to all three Enterobacteriaceae in this study, except α-folC, α-csgD, and α-fnr which are designed to be specific to *E. coli*, and α-recA which is specific to *E. coli* and STm. PNAs α-folC, α-ffh, α-lexA, α-acrA, α-recA, α-csgD, and α-fnr target novel pathways of metabolism, signal recognition, stress response, transport, stress response, biofilm formation, and metabolism respectively. PNAs α-gyrB and α-rpsD target novel genes in traditional antibiotic pathways. **b** Following PNA solid-phase synthesis, the product is purified using HPLC (representative chromatogram shown), verified by LCMS (representative spectra shown), and tested against a lab strain of *E. coli* (MG1655, growth curves for α-lexA and α-rpsD from Figure S5 shown). **c** Antibiotic resistance characterization of clinical isolates of CRE *E. coli*, MDR *E. coli*, ESBL KPN, NDM-1 KPN, and MDR *S.* Typhimurium. (left) Antibiotic resistance characterization of clinical isolates used in this study. Letters "R", "S", and "I" indicate drug-resistance, sensitivity, and intermediate resistance respectively. Sensitive, intermediate, and resistant breakpoints are provided in Table S1. MIC90 antibiotic concentration ranges for clinical isolates are provided in Table S2. Nine antibiotics of varied mechanisms and classes were tested including penicillins (ampicillin, AMP), cephalosporins (ceftriaxone, FRX), carbapenems (meropenem, MER), aminoglycosides (gentamicin, GEN and kanamycin, KAN), tetracyclines (tetracycline, TET), fluoroquinolones (ciprofloxacin, CIP), quinolones (nalidixic acid, NXA), and phenicols (chloramphenicol, CHL). **d** Clinical isolate monotherapy testing is done by monitoring growth at $OD_{600nm}$ over 16 h. Shown here are representative growth curves from Figure S7 of four of the clinical isolates with 10 μM PNA: NDM-1 KPN and a PNA control, α-nonsense, MDR *E. coli* and α-ffh, CRE *E. coli* and α-rpsD, and MDR STm and α-acrA with at least three biological replicates and errors bars as standard deviation. **e–f** Normalized growth (ratio of optical density of treatment to no treatment at 16 h) of clinical isolates in the presence of treatment with 10 μM of the indicated PNA. PNAs targeting essential and non-essential genes and showing at least 50% growth inhibition are shown in panels **e** and **f** respectively with significance (represented by an asterisk, *p* value <0.05) determined relative to control nonsense PNA and no treatment respectively. All data shown are the average of at least three biological replicates with standard deviation shown as error bars. Grey circles indicate individual biological replicates. **g** PNA concentration ranges for 90% growth inhibition of each bacteria strain where shaded grey boxes indicate expected PNA homology for the given bacterial strain.

transport protein, non-essential). We also selected PNA targets that were not from the 71 essential or 243 non-essential candidates with homology in all three MDR species to demonstrate the PNA's species selectivity. We selected one essential and two non-essential targets in non-traditional pathways specific to only *E. coli*: *folC*[37] (essential, H2 folate synthetase), *csgD*[38] (curli fimbriae expression, non-essential), and *fnr*[39] (anaerobic metabolism, non-essential). In addition, we chose to target *recA*[40] (SOS double stranded break repair, non-essential) because it shows homology to *E. coli* and *Salmonella* but not KPN.

The efficiency of the PNA Finder tool reduces the requisite time to obtain antisense antibiotic sequences, and the discovery period becomes limited merely by the duration of the FAST platform's synthesis, purification, and testing. Prior to testing on MDR clinical isolates, we selected two of this set of PNA, α-rpsD and α-lexA, to evaluate how quickly these latter steps of the FAST pipeline could be accomplished. It was determined that solid-phase synthesis, using Fmoc chemistry at a 10-micromole scale, of the 23-residue peptides (see Methods, Fig. S1) required less than three days. This was followed by purification using high performance liquid chromatography (HPLC), confirmation using liquid chromatography–mass spectrometry (LC–MS), and lyophilization; this process required less than 7 hours (Fig. 2b, S2, & S3). Testing required an additional 24-h experiment and validated the predicted toxicity of PNA against their essential gene targets in *E. coli* reference strain MG1655 (Fig. 2b, S4, & S5).

**MDR clinical isolates' antibiotic resistance characterization.** Prior to treatment of MDR clinical isolates with the PNA designed by the FAST platform, we characterized both the phenotypic and genotypic resistance profiles of the five MDR clinical isolates that were obtained. These isolates included a carbapenem-resistant Enterobacteriaceae (CRE) *E. coli*, an MDR *E. coli*, an extended spectrum β-lactamase (ESBL)-producing *K. pneumoniae* (KPN), a New Delhi Metallo β-lactamase 1 (NDM-1) KPN, and an MDR *S. enterica* serovar Typhimurium (STm). Phenotypic antibiotic resistance characterization of the clinical isolates was performed using nine antibiotics of varied mechanisms and classes, and comparing to 2016–2017 CLSI breakpoint

values[41] (Fig. 2c, Table S1). We found all isolates to have resistance to two or more antibiotics, with CRE *E. coli* showing resistance to all nine antibiotics tested (Table S2). Genome sequencing showed all of the clinical isolates to have at least two unique antibiotic resistance genes, and at least one β-lactamase gene (Fig. 2c, Table S3 & S4). The sequencing also allowed us to confirm the presence of each antisense target sequence in each isolate. In addition, Find Off-Targets was used to search these genome assemblies for off-targets around the start codon. Among all PNA and MDR strains we predicted only one PNA to have a single inhibitory off-target in each clinical isolate, α-rpsD in MDR STm and α-acrA in CRE *E. coli*, MDR *E. coli*, ESBL KPN, and NDM-1 KPN (Fig. S6, Table S5).

**PNA growth inhibition efficacy as a monotherapy.** Each PNA was conjugated to the cell-penetrating peptide (CPP) $(KFF)_3K$ to enhance transmembrane transport[22,23,42,43] and administered to clinical isolates at a 10 μM concentration, based on the minimum inhibitory concentration (MIC) observed in *E. coli* MG1655 (Fig. S4 & S5), to assess the toxicity of each as a monotherapy. Inhibition was compared to a scrambled sequence PNA (α-nonsense) as a control. Eighteen out of 34 treatments that showed homology to target clinical isolates demonstrated at least 50% growth inhibition, as shown in Fig. 2d–f (Fig. S7). Further, minimum inhibitory concentration analysis for 90% growth inhibition (MIC90) showed six treatments exhibit MIC90s below 10 μM (Fig. 2g, Fig. S8). The PNA α-rpsD was found to be the most successful monotherapy, causing a greater than 50% growth reduction in all five clinical isolates and all MIC90 values below 15 μM (Fig. 2d, e, & g, Fig. S7 & 8). α-gyrB, α-lexA, and α-ffh were found to significantly inhibit growth in all clinical isolates but NDM-1 KPN (*p* vale < 0.05). Of the PNAs targeted to only *E. coli*, α-folC (Fig. 2e), and α-csgD (Fig. 2f), successfully inhibited growth (*p* value < 0.05) in both *E. coli* isolates, whereas α-fnr (Fig. S7) was effective against only the CRE *E. coli* strain. α-recA, which is specific to STm and *E. coli*, was effective against both *E. coli* strains and MDR STm. PNA treatments demonstrated high selectivity to their target strains. Of the 45 total treatments, 34 were predicted to have homology to the clinical isolate tested and eleven did not. Seven out of eleven PNA treatments without

sequence homology to the clinical isolate showed no growth reduction; the remaining four showed less than 10% growth inhibition (Fig. S7 & S8). Of the 34 clinical isolate-PNA pairs predicated to have homology 28 showed significant growth inhibition, with six monotherapy treatments reducing growth by more than 97% (Fig. 2e, Fig. S7 & S8). This success rate is remarkable given the high difficulty in treating these strains with conventional antibiotic strategies.

In analyzing these results, we looked for correlations between growth inhibition and each gene target's corresponding mRNA abundance level, secondary structure, and protein network interactions (Fig. S9, S10). Though a trend has been suggested in previous antisense antibiotic research[44], we did not identify any correlation between predicted secondary structure and growth inhibition, nor did we identify a correlation with mRNA abundance. We observed a significant correlation (Pearson correlation coefficient $r = 0.908$, $p$ value $< 0.001$) between growth inhibition and protein interaction network connectedness denoted by average node degree—as identified by STRING protein network analysis[31] (Fig. 3a). The positive correlation indicates that the more interconnected a network is the more susceptible it is to PNA treatment. Conversely, loosely connected protein interaction networks (low node degree) showed weaker monotherapy effects.

**Antibiotic and PNA combination therapy treatment on resistant isolates at close to CLSI "sensitive" breakpoints.** We tested PNAs that showed minimal to moderate growth inhibition in monotherapy treatment in combination with small molecule antibiotics to which the MDR clinical isolates had demonstrated resistance (Fig. 2c, Fig. 3b-p). Treatment interaction was evaluated using the Bliss Independence model[45], with positive $S$ values indicating synergy or greater inhibition than predicted by a simple additive effect. In CRE *E. coli* we identified significant synergistic interaction between α-gyrB (10 μM, essential), and both chloramphenicol (8 μg/mL, Fig. 3b & d, Fig. S11), and gentamicin (4 μg/mL, Fig. 3e, Fig. S11) with $S$ values of $0.64 ± 0.09$ and $0.56 ± 0.12$, respectively. In addition, the non-essential PNA targets of α-acra, α-csgD, and α-recA all showed significant synergistic interaction with gentamicin (4 μg/mL, 10 μM PNA, Fig. 3f-i, Fig. S11). The CRE *E. coli* isolate possesses resistance genes specific to the mechanisms of each antibiotic (Fig. 2c, Table S3 & S4), yet it was still possible to resensitize it to the antibiotics at their CLSI breakpoints (Table S1). We also identified significant synergistic interaction in ESBL KPN between the PNAs α-ffh, α-lexA, and α-gyrB (10 μM) and tetracycline (2 μg/mL, Fig. 3j-m, Fig. S11). Beyond single antibiotic and PNA combination treatments we performed a checkerboard analysis of synergy, which generally showed an optimal combination for synergy at 10 μM PNA and close to the CLSI "sensitive" breakpoint antibiotic concentration (Fig. 3n-p, Fig. S12-15), with only one deviation where low concentrations of MER and 10 μM α-rpsD showed significant antagonism in NDM-1 KPN (Fig. S13 & 15). In addition, we compared the synergy between chloramphenicol and α-gyrB with chloramphenicol and an antibiotic targeting the same enzyme affected by α-gyrB: ciprofloxacin. α-gyrB directs the synthesis of DNA gyrase subunit B and ciprofloxacin also acts on DNA gyrase. A checkerboard analysis showed no significant synergy between chloramphenicol and ciprofloxacin even though there was synergy between α-gyrB and chloramphenicol (Fig. S16). α-gyrB also showed synergy with tetracycline but was not furthered explored here.

Our data showed a negative correlation ($r = -0.3017$, $p = 0.0314$, Fig. 3q) between the degree of synergy and average node degree, indicating that targeting genes that have less interconnected protein interaction networks could give rise to synergy with antibiotics. This is likely due to the fact that when targeted together the result is perturbations of a larger network which extends the range of effects. In contrast, targeting a protein with more network connections naturally has a stronger therapeutic effect by itself, which is not improved upon by synergy.

Further, in contrast with previous research that showed a higher degree of antibiotic synergy between antisense oligonucleotides and small molecule drugs when the treatments targeted related pathways[46,47], our results indicate that unrelated mechanisms of inhibition can also result in combination effects in MDR pathogens[27]. This phenomena has also been shown by our previous works highlighting how multiplexed perturbations have a compounding effect on bacterial fitness[48] and that fitness-neutral gene perturbations can have synergistic effects across similar or different pathways[49]. Our data demonstrate that the FAST platform is capable of designing PNAs that can be highly effective both as a monotherapy as well as potentiators of traditional antibiotics in drug-resistant pathogens. Further, FAST provides insights into the role protein interaction networks have in PNA's therapeutic activity and provides an approach to expand both the utility of the platform and the effective supply of usable antibiotics against MDR pathogens.

**PNA delivery into infected mammalian cells using the T3SS.** The final obstacle for PNA antibiotic viability that the FAST platform seeks to address is the molecules' poor uptake into mammalian cells to treat mammalian intracellular infections[50,51]. Though conjugation to a CPP improves PNA transmembrane transport in bacteria, transport into mammalian cells still remains limited, creating challenges for in vivo application of the therapy. Though nanoparticle and lipoparticle based approaches are being developed, these have challenges of poor tissue targeting and lower stability[50]. To overcome some of these limitations, here we present a probiotic-based delivery approach for delivering antisense oligomers to infected mammalian cells. In the delivery phase of the FAST platform, we present a first probiotic prototype for PNA transport using the Type III secretion system (T3SS), a gram-negative bacterial machine for invasion of the eukaryotic host cell[52,53], combined with a lysis switch to release the therapeutic.

First, we sought to establish that a strain expressing the T3SS was capable of transporting PNA—α-rpsD and α-lexA were chosen for this evaluation—into human epithelial cells (HeLa). We confirmed that PNAs are not lethal to mammalian cells in the range tested (Fig. S17). To confirm transport into mammalian cells we used the *S. typhimurium* strain SL1344, which expresses both the T3SS and green fluorescent protein (GFP)[54] (the strain is referred to from here as STm-GFP). We examined transport effectiveness of PNA delivered into HeLa cells via the bacteria's T3SS-mediated infection (T3SS-PNA conditions) compared to PNA added extracellularly post T3SS infection (naked PNA conditions) by measuring intracellular STm-GFP clearance (Fig. 4a). For both conditions HeLa cells were infected with STm-GFP at a multiplicity of infection (MOI) of 10 for 45 min. For the T3SS-PNA conditions STm-GFP were exposed to 10 μM α-rpsD or α-lexA PNA during the 45-min infection. During this 45-min time period the PNA was allowed to enter the STm-GFP, an incubation time shown to allow for reasonable PNA uptake[51,55,56], as it infected the HeLa cells, but this time period was not long enough for PNA to decrease STm-GFP viability (Fig. S18). After 45 min of infection, the cells of both conditions (T3SS-PNA and naked PNA) were washed and incubated in gentamicin-containing

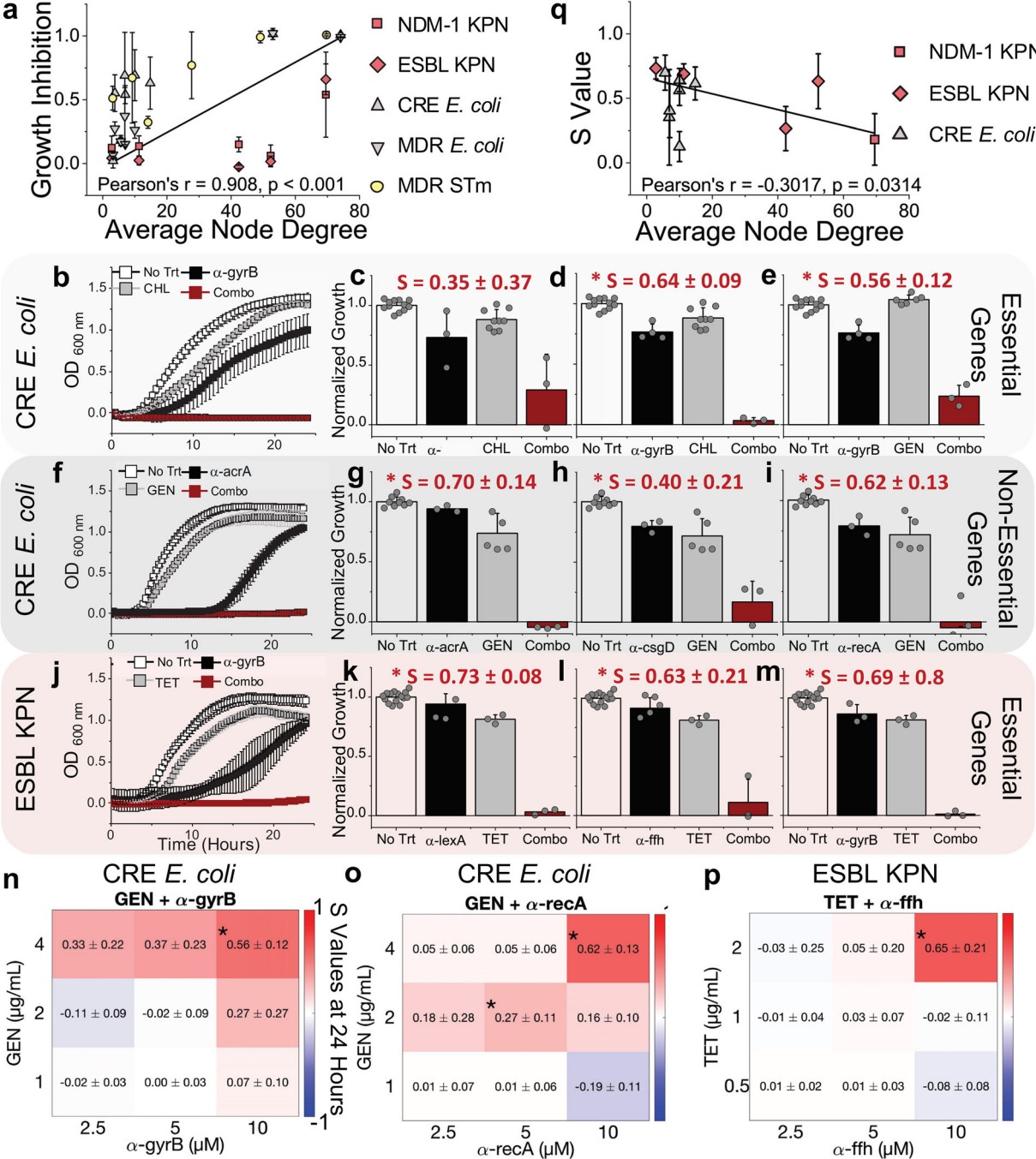

**Fig. 3 FAST platform generates PNAs that can potentiate activity of traditional small molecule antibiotics. a** Using the STRING database of protein network interactions (data shown in Supplementary Figure S10), we found a positive correlation, Pearson's $r = 0.908$ and $p < 0.001$, between growth inhibition of PNA homologous with target bacteria and average node degree, the number of interactions of protein has in the average network. **b–m** Subsequent growth curves and bar plots show bacteria without treatment, or treatment with PNA alone (10 μM), or antibiotic alone, or PNA and antibiotic combined. Antibiotic concentrations were (from top to bottom) 2 μg/mL tetracycline (TET), 4 μg/mL gentamicin (GEN), and 8 μg/mL chloramphenicol (CHL). **b, f, j** The first column shows representative growth curves over 24 h of each treatment. Panels **b–e** and **f–i** show treatment of CRE *E. coli* with PNAs targeting essential and non-essential genes respectively. Panels **j–m** show treatment of ESBL KPN with PNAs targeting essential genes. All bar plots are the average OD (600 nm) of each treatment at 24 h normalized to no treatment at 24 h. *S* values above the bar plots were obtained using the Bliss Independence model and indicate synergistic interaction between PNA and antibiotic at 24 h, an asterix indicates significance at $\alpha = 0.05$. Grey circles indicate individual biological replicates. Panels **n–p** show select heat maps of synergy values at 24 h of checkerboard combination assays (Figures S12-13). **q** Plotting the *S* values representing synergy against the target protein's average node degree shows a negative correlation (Pearson's $r = -0.3017$, $p = 0.0314$). All data shown are the average of at least three biological replicates with standard deviation shown as error bars.

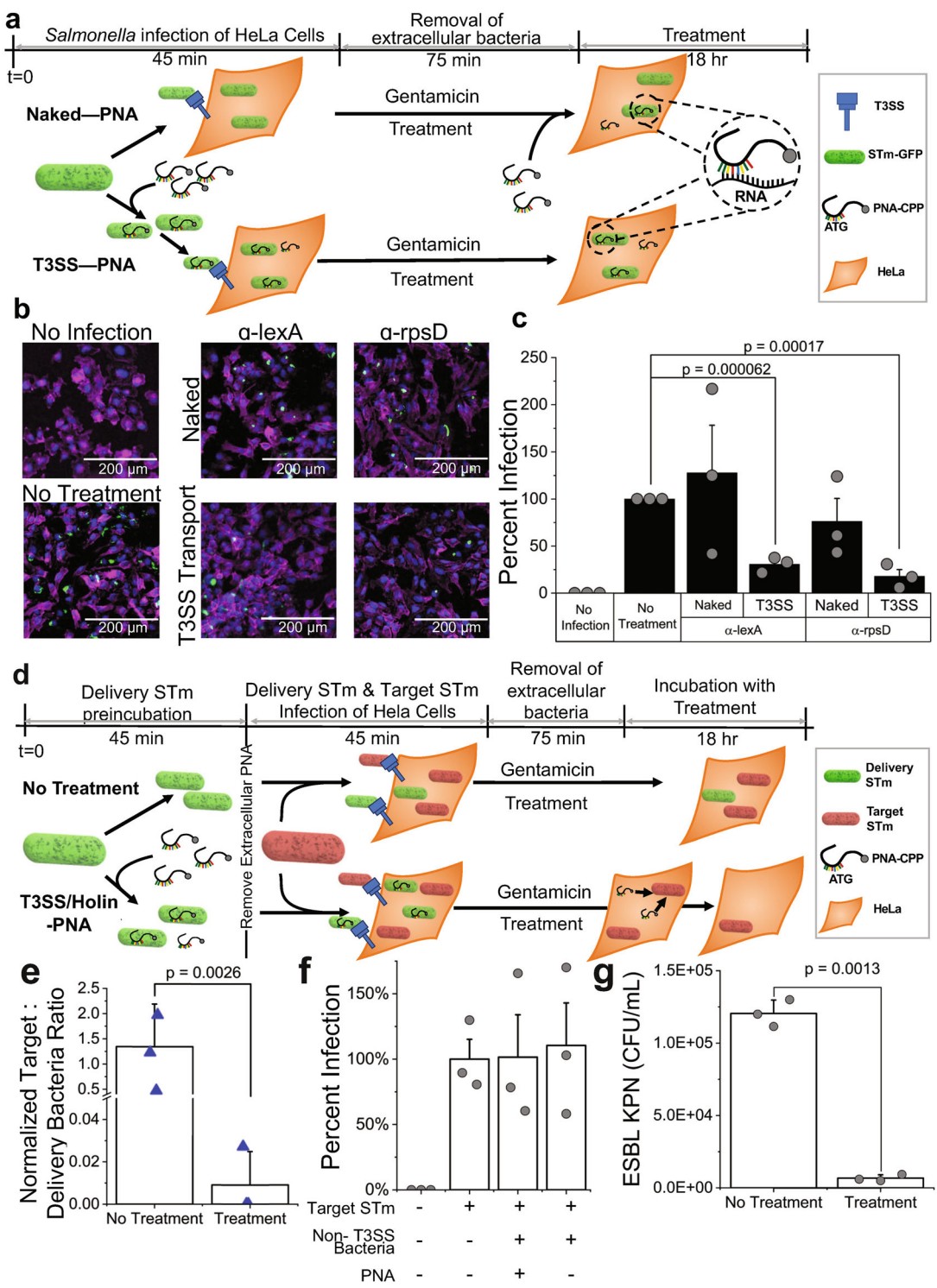

media to remove any remaining extracellular bacteria and then incubated for a period of 18 h. For the naked-PNA treatment PNA was only added during this 18-h incubation period. Both α-rpsD and α-lexA inhibited STm-GFP intracellular growth by more than 60% in the T3SS-PNA conditions (Fig. 4b & c, Fig. S19), whereas neither reduced infection in the naked PNA condition even though PNA was added extracellularly for 18 h as compared to the 45-min co-incubation. The same experiment was repeated in a different mammalian cell line, murine osteoblasts, with reproducible results indicating it is not a unique phenomenon to a HeLa intracellular infection (Fig.

S20). These results demonstrate that *S.* Typhimurium, expressing the T3SS, can act as a delivery vessel of PNA into human epithelial cells.

### PNA treatment of infected mammalian cells using the T3SS delivery and holin–endolysin release

We next engineered the STm-GFP strain to release PNA into the mammalian cell to target a separate intracellular pathogen. We constructed a modified strain of STm-GFP in which cell lysis can be induced by a holin–endolysin system[57] under the control of a laciQ promoter

**Fig. 4 Repurposing the bacterial Type III secretion system (T3SS) for PNA delivery and release to clear intracellular infections of mammalian cells.**
**a** Schematic showing the experimental setup for comparing PNA delivery into HeLa cells for treating an intracellular infection of *Salmonella enterica* serovar Typhimurium, strain SL1344 expressing GFP (STm-GFP), using either the bacterial T3SS inherent to STm-GFP (Bottom, T3SS-PNA) or CPP facilitated uptake (Top, Naked-PNA). All intracellular infection treatment experiments follow the general protocol of 45-min infection of HeLa cells 24 h after seeding, 75-min gentamicin treatment to remove extracellular bacteria, and 18-h incubation with treatment in media containing gentamicin to ensure an only intracellular infection model. For comparing T3SS delivered PNA to naked-PNA infection was done using STm-GFP at an MOI of 10. For the T3SS-PNA condition 10 μM of PNA was added during this 45-min infection stage; during this time the PNA is allowed to enter the *Salmonella* and subsequently be transported into the HeLa cell as it infects via its T3SS. For the naked-PNA condition 10 μM of PNA is added to the gentamicin containing media that the HeLa cells are incubated in for 18 h. Post 18 h of treatment HeLa cells are fixed for imaging or lysed for colony forming unit (CFU) analysis. **b** Representative images of HeLa cells uninfected (top left), infected and without treatment (bottom left), infected and treated with Naked-PNA (top right), and infected and treated with T3SS-PNA (bottom right). HeLa cells are stained with the nuclear stain DAPI (blue), the membrane stain Phalloidin (pink), and green pixels represent intracellular STm-GFP (green). Images show an evident decrease in STm-GFP during T3SS-α-rpsD or T3SS-α-lexA treatment compared to no treatment or with naked-PNA treatment. **c** Percent infection (CFU/mL of treatment normalized to no treatment) in the presence or absence of treatment. Significant reduction in STm in HeLa cells when PNA is delivered using the T3SS. Naked-PNA treatment does not produce significant therapeutic effect. **d** Schematic showing the experimental setup comparing the delivery and release approach to no treatment of an intracellular infection of HeLa cells. The same procedure was followed as in part A with few modifications. Pretreatment of the delivery STm (STm-GFP modified with holin release switch) is done by incubating in either 10 μM of PNA (bottom, T3SS/holin-PNA) or PBS (top, no treatment) for 45 min to allow for PNA uptake. Then HeLa cells are infected with a 4:1 ratio of delivery STm (green) to target STm (red), making the total MOI 10. After removing extracellular bacteria, the infected HeLa cells are incubated for 18 h; during this time the delivery STm releases the PNA to treat the Target STm strain. After 18-h of incubation the HeLa cells are lysed to determine the number of intracellular colony forming units. **e** CFU ratio of normalized target STm to normalized delivery STm (see methods section for normalization) shows significant reduction compared to no treatment. Triangles represent individual biological replicates of HeLa cells. **f** Percent infection of target STm (Target STm CFU normalized to no treatment) when using a non-T3SS bacteria (*E. coli* MG1655) as the delivery strain shows no reduction ($p > 0.05$). **g** Lysed ESBL KPN (CFU/mL) from intracellular infection in RAW264.7 macrophages at 1 MOI and treated for 18 h with delivery STm at 10 MOI carrying α-rpsD. All data shown are the average of three biological replicates with standard deviation shown as error bars. Grey circles or blue triangles indicate individual biological replicates.

(strain referred to as delivery STm, Fig. S21) to treat a separate intracellular pathogen, SL1344 expressing mCherry via a plasmid (strain referred to as Target STm). This promoter had a lac operator downstream—shown to permit RNA polymerase read-through even for tightly bound LacI proteins[58]—which enabled basal expression sufficient to cause holin-induced lysis in the absence of IPTG induction (Fig. S21B). To demonstrate assay generalizability, we used a different PNA, α-recA, which was found to be lethal to STm-GFP at 10 μM (Fig. S22). To treat a separate intracellular infection, we allowed a 45-min incubation period of the delivery STm—with either 10 μM α-recA (T3SS/holin-PNA condition) or PBS (no treatment). The target STm was incubated with PBS in parallel at equal cell densities. Delivery STm (with or without PNA) was washed to remove any extracellular PNA and was then co-incubated with the target STm and HeLa cells, at a 4:1 ratio of delivery to target STm to achieve a combined MOI of 10. The cultures were then washed again and incubated in gentamicin-containing media for 18 h to remove extracellular bacteria (Fig. 4d). Growth inhibition was measured by comparing the ratio of bacterial load after to before infection, to account for any variability in initial bacterial load used for infection across replicates. In the T3SS/holin-PNA condition the intracellular load of the target STm, normalized to the delivery STm load, was reduced by more than 99% compared to the no PNA treatment condition. For two out of three biological replicates, the target strain was completely eliminated across a range of MOI (Fig. 4e, Fig. S23 & 24). In contrast, when using a non-T3SS bacteria, *E. coli* MG1655, as the delivery strain, there was no decrease in the target STm load (Fig. 4f). We further validated the delivery STm's efficacy by using it to treat an intracellular ESBL KPN infection in RAW264.7 macrophages. Infection and treatment were done similarly to the HeLa double infection study with a few modifications. Macrophages were infected with 1 MOI ESBL KPN and 10 MOI delivery STm (+/- treatment of α-rpsD) at the same time in 50% DPBS and 50% DMEM containing 10% FBS and synchronized by centrifugation. Following gentamicin protection, 18 h incubation, and lysis, ESBL KPN colony forming units were selectively enumerated on kanamycin containing LB

plates, delivery STm is not kanamycin resistant. Figure 4g shows a significant drop in ESBL KPN infection with treatment compared to no treatment, validating the delivery of PNA by the type 3 secretion system in a third mammalian cell line while also treating a secondary infection of a different bacteria. These data provide evidence that a combination of the T3SS with a holin–endolysin system is an effective solution for the delivery of PNA to treat intracellular infections, which remains a major hurdle in antisense antibiotic treatments. Though *S. typhimurium* is a pathogenic species, future work towards the expression of the T3SS and holin–endolysin system within a probiotic species will enable the efficient and safe transport of PNA into mammalian cells.

We have demonstrated the utility of the FAST platform to quickly and effectively create PNA antibiotics, through the stages of design, synthesis, testing, and delivery. Development of the FAST platform will help to create a much needed rapid and adaptive countermeasure platform for accelerating the response to emergence of multidrug resistance. The PNA Finder toolbox is capable of designing and screening hundreds of PNA candidates simultaneously, in order to narrow down a list of sequences that specifically target a harmful pathogen while avoiding targeting other species. The PNA Finder toolbox is a dynamic component and, as more PNA antibiotics are tested, will incorporate feedback into its algorithm to improve the PNA target suggestion list based on weighted parameters correlated with the PNA's predicted efficacy. The quick design process allows for the creation and testing of potential antibiotic therapies in under a week, and our results have demonstrated that this rational design process results in much higher monotherapy success rates than conventional small molecule screening pipelines[59]. In addition, we demonstrate the utility of the FAST platform to create antibiotics that are effective potentiators of conventional antibiotic activity, even in strains with specifically identified resistance to small molecule drugs. Finally, we have engineered a hybrid type III secretion/ holin–endolysin system for effective therapeutic intracellular delivery. The results presented can help develop a novel probiotic PNA delivery method and improve FAST platform's relevance for

clinical applications. The application of the FAST platform to design of pathogen- and gene-specific antibiotics in non-traditional antibiotic pathways could be instrumental in the development of new antibiotics for already pervasive MDR pathogens.

## Methods

**PNA Finder toolbox**. The PNA Finder toolbox is built using Python 3.7, as well as the alignment program Bowtie 2 (version 2.3.5.1)[28], the read alignment processing program SAMtools (1.9)[29], and the feature analysis program BEDTools (v2.25.0)[30]. In addition, in order to run on a Windows operating system, the toolbox requires a Window-compatible bash shell (the program Cygwin has been used in development) to provide a Unix-like environment in which Bowtie 2, SAMtools, and BEDTools can be compiled and run. Mac operating systems are not currently supported by the toolbox. The user interface for the PNA Finder Toolbox is constructed using the Python 3.7 package Tkinter, version 8.6.

**Gene target selection**. Essential genes were identified using the Keio collection entry in the Database of Essential Genes (DEG)[60]. PNA sequences were identified using lists of gene IDs from DEG as inputs for the custom Python script, which uses a genome annotation and associated assembly to extract the reverse complements of 12-mer nucleotide sequences centered on the mRNA AUG start codons (STC) for the genes of interest. Using the Find Off-Targets function of the PNA Finder toolbox, these sequences were screened for off-target alignments within the genome assembly for *E. coli* MG1655 (RefSeq: GCF_000005845.2), from which the PNA sequences originated, as well as for targeting homology within the genome assemblies for KPN MGH 78578 (RefSeq: GCF_000016305.1) and STm SL1344 (RefSeq assembly: GCF_000210855.2). PNA-RNA complements were predicted to cause inhibition if they were located within 20 bases of the mRNA start codon, based on previous work[26]. The Get Sequences function of the PNA Finder toolbox was used to search for indicators of low solubility as described by Gildea et al[61], such as purine stretches greater than five bases, a purine content of greater than 50%, or a guanine–peptide stretch of longer than three bases. The function also searched for self-complementary sequences of greater than five bases. PNA target pathways were identified using EcoCyc[62] and Regulon Database[63].

**PNA Synthesis**. All PNAs contained sequences complementary to target mRNA, as well as the CPP (KFF)$_3$K attached via an AEEA or O linker. These sequences are provided in Fig. S1. PNA for the toxicity studies (Fig. 2b, growth curve, Fig. S7 α-nonsense in first row, Fig. S8 & Fig. S12-15) were synthesized using an Apex 396 peptide synthesizer (AAPPTec, LLC) to perform solid-state PNA synthesis using Fmoc chemistry on MBHA rink amide resin at a 10 μmol scale. Fmoc-PNA monomers were obtained from PolyOrg Inc. A, C, and G monomers are protected at amines with Bhoc groups. All PNAs were synthesized with a CPP (KFF)$_3$K, which has lysine residues protected with Boc groups. PNA products were precipitated and purified as trifluoroacetic acid salts. PNAs used in MDR clinical isolate growth experiments were ordered from PNA Bio Inc. (Newbury Park, CA), and were also conjugated to the cell-penetrating peptide (KFF)$_3$K. PNA used for the toxicity measurements in mammalian cells was a generous gift provided by Prof. Teruna Siahaan's lab at the University of Kansas (Fig. S17). PNA was dissolved in H$_2$O with 5% DMSO at 100 μM. Stocks were stored at −20 °C for long-term and at 4 °C for working stocks to minimize freeze/thaw cycles.

**Synthesis of high concentration PNA for toxicity measurements**. PNA (α-lexA and α-nonsense) used for HeLa high concentration toxicity measurements (Fig. S17) were synthesized using solid-phase chemistry on Fmoc-MBHA resin (0.22 mmol/g loading) at a 0.02 millimolar scale. Fmoc deprotection steps were performed using piperidine and coupling was performed using HATU as an activator. Post coupling, acetylation was carried out with a 5%/6% v/v solution of acetic anhydride and 2,6-lutidine, respectively, in DMF. Cleavage was performed for 2 h using an 88%/2%/5%/5% v/v solution of TFA, TIPS, phenol, and water, respectively. The PNA molecule was precipitated in ethyl ether, centrifuged, and the purified using high pressure liquid chromatography.

**Gel shift mobility assay**. Synthetic DNA oligonucleotides (Table S6) that are 57–60 nucleotides in length were incubated with their respective PNA binding site at 37 °C overnight. The concentration of the DNA fragments was held at 500 nM, while the PNA was always in excess at 1 μM. The reactions were performed in 1X TE with 20 mM KCl (pH 7.0)[64–66]. The formation of PNA-ssDNA complexes was observed on a 20% polyacrylamide nondenaturing gel using 1X TBE running buffer. 1X SYBR-Gold® was used to stain and visualize the DNA using about 50 mL to coat the gel in low-light conditions for 20 min. The gel was directly put on a UV sample tray and imaged using the Gel Doc™ EZ Imaging system from Bio-Rad (Fig. S6E).

**Bacterial cell culture**. The clinical isolates were obtained from the lab of Nancy Madinger at the University of Colorado Anschutz campus. Clinical isolates were grown in Cation Adjusted Mueller–Hinton broth (CAMHB) (Becton, Dickinson and Company 212322) at 37 °C with 225 rpm shaking or on solid CAMHB with 1.5% agar at 37 °C. Clinical isolates were maintained as freezer stocks in 90% CAMHB, 40% glycerol at −80 °C. Freezer stocks were streaked out onto solid CAMHB and incubated for 16 h to produce single colonies prior to experiments. For each biological replicate, a single colony was picked from solid media and grown for 16 h in liquid CAMHB prior to experiments. Non-clinical isolates, such as *E. coli* MG1655 (ATCC700926) and *Salmonella enterica* serovar Typhimurium SL1344, expressing GFP from the chromosome (*rpsm*::gfp)[54], were cultured in liquid 2% lysogeny broth (LB) or on 2% LB with 1.5% agar for solid plates. SL1344 cultures were supplemented with 30 μg/mL streptomycin and SL1344 strains containing plasmids pFPV (target STm) or pRG1 modified with a lacIq promotor (SL1344-holin) were supplemented with 100 μg/mL ampicillin. Freezer stocks were stored in 60% LB broth, 40% glycerol at −80 °C. Freezer stocks were streaked out onto solid LB and incubated for 16 h to produce single colonies before experiments. For biological replicates, single colonies were started in liquid LB and grown for 16 h prior to experiments. *E. coli* MG1655 PNA growth experiments were carried out in M9 media (1x M9 minimal media salts solution (MP Biomedicals), 2.0 mM MgSO$_4$, and 0.1 mM CaCl$_2$ in sterile water) with 0.4% glucose.

**Antibiotic resistance screening**. Sensitive/resistant breakpoints were taken from the 2016–2017 Clinical & Laboratory Standards Institute report[41] (Table S1). Liquid cultures of the clinical strains were diluted to a 0.5 McFarland standard and added to the respective antibiotic test condition. The antibiotic minimum inhibitory concentration (MIC) for each clinical isolate was determined as the lowest antibiotic concentration which prevented visible cell growth for 24 h. Strains were "sensitive" if the MIC was equal to or below the sensitive-breakpoint concentration, "resistant" if the MIC was greater than or equal to the resistant-breakpoint concentration, and "intermediate" if the MIC was in-between.

**Genome sequencing library prep and data analysis**. Liquid cultures were inoculated from individual colonies off of solid cation-adjusted Mueller–Hinton broth (CAMHB) for each clinical isolate. Cultures were grown for 16 h as described above and then 1 mL of culture was used to isolate DNA using the Wizard DNA Purification Kit (Promega). A Nanodrop 2000 (Thermo Scientific) was used to measure DNA concentration and purity. For library preparation, >2 μg of DNA was submitted in 50–100 μL samples. The libraries were prepared for sequencing using Nextera XT DNA Library Preparation Kit (Illumina), and the sequencing was run with a 2 × 250 bp MiSeq run (Illumina).

Sequencing reads were first trimmed using TRIMMOMATIC v0.32[67] for length and quality with a sliding window. For further analysis, the trimmed files, of only paired sequences, were transferred to Illumina BaseSpace (http://basespace.illumina.com). We assessed the sequencing quality using FASTQC v1.0.0 and performed de novo genome assembly with SPAdes Genome Assembler v3.6.0[68,69]. The assembly was further corrected and improved using Rescaf v 1.0.1, and then we performed annotation using PROKKA v 1.0.0[70]. Antibiotic resistance genes were identified and characterized using SEAR and ARG-ANNOT pipelines[71,72]. Integrated Genomics Viewer[73] was used for data visualization.

**Homology analysis for PNA in clinical isolates**. The PNA Finder Find Off-Targets function was used to identify homologous nucleotide sequences and determine whether PNA complementation at these loci would be predicted to cause translation inhibition. For *S. typhimurium* and *K. pneumoniae* strains, annotations were examined manually to determine whether the PNA would be expected to inhibit a gene analogous to its original *E. coli* target.

**PNA growth assays**. Biological replicates were diluted 1:10,000 into treatment condition in 384-well plates and measured for 24 h in a Tecan GENios at 562 nm with a bandwidth of 35 nm. Media absorbance blanks were subtracted from data prior to analysis. Normalized optical density (OD) data is shown normalized to the "no treatment" growth curves at 16 h.

**Potentiation of antibiotics with PNAs**. Combination growth curve experiments were performed following the same procedure as the PNA growth assay mentioned above, except for the addition of antibiotics where appropriate.

Combinatorial effects were evaluated using the Bliss Independence model[74] where the S parameter defines a deviation from no interaction as is defined as:

$$S = \left(\frac{OD_{AB}}{OD_0}\right)\left(\frac{OD_{PNA}}{OD_0}\right) - \left(\frac{OD_{AB,PNA}}{OD_0}\right) \tag{1}$$

Where $OD_{AB}$ is the OD at 24 h in only antibiotic, $OD_0$ is the OD at 24 h without treatment, $OD_{PNA}$ is the OD at 24 h in only antisense-PNA at 10 μM, and $OD_{AB,PNA}$ is the OD at 24 h in a combination of antibiotic and antisense-PNA. $S > 0$ is a deviation towards synergy and $S < 0$ is a deviation towards antagonism. S value significance was determined using a one-sample *t* test with a hypothesized mean of zero and individual errors propagated.

**Plasmid and strain construction**. The SL1344-holin strain contains a modified pRG1 plasmid that is IPTG inducible. The original pRG1 plasmid expresses the SRRz genes of the lysis cassette[75]. The lacIq gene was inserted in between the SalI and BamHI cut sites. The lacIq gene was extracted from the *E. coli* strain DH5αz1 by colony PCR with Phusion High-Fidelity DNA Polymerase (New England Biolabs) and with the following primers: forward primer (5′-AAAGGATCCCAT-CACTGCCCGCTTTCCAGTCG-3′) and reverse primer (5′-AAAGTCGACCCGACACCATCGAATGGTGCAAAACCTTTCG-3′). The PCR products were subsequently gel-purified (Zymoclean Gel DNA Recovery Kit, Zymo Research Corporation), digested sequentially with *BamHI* and *SalI* (FastDigest Enzymes, Thermo Scientific) as per provided protocols, and PCR-purified (GeneJET PCR Purification Kit, Thermo Scientific) between and after digestion. The pRG1 backbone was also digested with *BamHI* and *SalI* and gel purified, and T4 DNA Ligase (Thermo Scientific) was used to ligate the pRG1 backbone and the extracted lacIq gene. Ligations were transformed into electrocompetent SL1344 cells and plasmid minipreps were performed using Zyppy Plasmid Miniprep Kit (Zymo Research). Confirmation was done by measuring the optical density with and without 1 mM IPTG (Fig. S21). The SL1344-mCherry bacterial strain contains the pFPV-mCherry plasmid (Addgene #20956) which was transformed into electrocompetent SL1344 cells and confirmed by fluorescence when excited with light at 587 nm in a light box and viewed through a 610 nm emissions filter.

**Optical density growth measurements of SL1344-holin**. Single colonies of each bacterial strain were picked and grown in LB supplemented with 30 μg/mL streptomycin and 100 μg/mL ampicillin. Overnight cultures were diluted 1:1,000 into 100 μL of LB supplemented with 30 μg/mL streptomycin and 100 μg/mL Ampicillin and 1 mM IPTG where appropriate in a 96-well plate. Cultures were grown at 37 °C, with shaking, for 24 h and $OD_{600nm}$ measured every 30 min using a GENios plate reader (Tecan Group Ltd.) operating under Magellan software (version 7.2).

**String database analysis**. Gene names for each of the PNA-targeted RNA sequences were entered into the STRING database and searched in the organisms of *Escherichia coli* K12 MG1655, *Klebsiella pneumoniae* subspecies *pneumoniae* MGH 78578, and *Salmonella enterica* subspecies *enterica* serovar *Typhimurium* strain LT2. The meaning of network edges was set to confidence, and all active interaction sources were used except for text mining. The minimum interaction score was left at 0.400, while the max number of interactions was set to 250 for the 1st shell and zero for the 2nd shell. Counts of nodes and clusters were taken from the analysis tab.

**HeLa and RAW264.7 Cell Culture**. HeLa and RAW264.7 cells were stored as freezer stocks in 10% dimethyl sulfoxide (DMSO, Sigma) plus full media: Dulbecco's Modified Eagle Medium (DMEM, Fisher Scientific), 10% Fetal Bovine Serum (FBS, Advanced, Atlanta Biologics), and 50 units/mL penicillin–streptomycin (P/S; Fisher Scientific). For biological replicates a single freezer stock, passage two, was split into three different culture flasks and each flask continuously passaged as individual biological replicates in full media. Cells were cultured at 37 °C, 5% CO₂, and controlled humidity. Cells were passaged at 80% confluency with 0.25% trypsin (HyClone). For infection experiments, HeLa and RAW264.7 cells were seeded in 96-well tissue culture treated plates (Fisher Scientific) at 45,000 cells/mL and 500,000 cells/mL respectively in 100 μL per well 24 h prior to infection.

**Osteoblast precursor cell culture**. MC3T3-E1 osteoblast precursor cells were stored in liquid nitrogen in growth media supplemented with 10% dimethyl sulfoxide (DMSO, Sigma). Osteoblasts were recovered from freezer stocks and cultured in growth media consisting of α-Minimum Essential Media (α-MEM, Gibco) supplemented with 10% Fetal Bovine Serum (FBS, Advanced, Atlanta Biologics), and 50 units/mL penicillin–treptomycin (P/S, Fisher Scientific) at 37 °C, 5% CO₂, and controlled humidity. For biological replicates a freezer stock, at passage 9, was split three ways and each replicate continuously passaged as individual biological replicates at 80% confluency using 0.25% trypsin (HyClone). For toxicity experiments osteoblasts, between passages 10 and 15, were seeded on to 96-well tissue culture treated plates (Fisher Scientific) at 100,000 cells/mL in 100 μL growth media for 24 h before infection or treatment.

**Osteoblast precursor infection and toxicity measurement**. Osteoblast cells were infected by replacing the growth media with Dulbecco's phosphate-buffered saline containing *Salmonella* at a concentration equivalent to a multiplicity of infection of 30 and incubated at growth conditions for 45 min. After infection media was replaced with growth media supplemented with 30 μg/mL gentamicin instead of penicillin–streptomycin, incubated for 75 min, and replaced with fresh gentamicin containing media and treatment conditions (150 μL per well). After 18 h of treatment 90 μL of supernatant was used to determine lactate dehydrogenase (LDH) release as a measure of cytotoxicity using the CytoSelect™ LDH Cytotoxicity

Assay Kit. Percent toxicity is defined and measured as:

$$Percent\ Toxicity = \frac{\left(Abs_{450\ nm}^{Treatment} - Abs_{450\ nm}^{Negative\ Control}\right)}{\left(Abs_{450\ nm}^{Positive\ Control} - Abs_{450\ nm}^{Negative\ Contorl}\right)} \quad \text{(S1)}$$

**Salmonella infection of HeLa cells and PNA treatment**. For a single infection (Fig. 4a-c), SL1344 cultures were inoculated from a single colony on solid media in to 1 mL of LB media with 30 μg/mL streptomycin and grown for 16 h at 37 °C with shaking. Cultures were diluted 1:10 and regrown in LB containing streptomycin for 3 h prior to infection. Bacteria were washed thrice with PBS, optical density at 600 nm measured, and diluted to a multiplicity of infection (MOI) of 10 in Dulbecco PBS (DPBS). MOI is calculated based on an established optical density and colony forming unit (CFU) calibration, and the number of mammalian cells per well at the time of infection approximated based on their 24-h doubling time. Mammalian cells were washed thrice with Dulbecco's phosphate-buffered saline (DPBS, Fisher Scientific) prior to addition of DPBS containing bacteria for a 45-min infection at HeLa growth conditions. After infection media was replaced with HeLa full growth media containing 100 μg/mL gentamicin instead of P/S to remove any extracellular bacteria and incubated for 75 min before replacement with full cell culture media containing 10 μg/mL gentamicin instead of P/S and respective treatment and incubated for 18 h.

Following 18 h treatment, wells for CFU analysis were washed thrice with 300 μL DPBS and lysed with 30 μL of 0.1% Triton X-100 for 15 min at room temperature. After 15 min, 270 μL of PBS was added to each well then serially diluted and plated on solid LB media supplemented with 40 μg/mL streptomycin. Plates were incubated for 16 h at 37 °C and CFU per milliliter determined. Wells for staining and imaging were fixed with 4% methanol-free paraformaldehyde at room temperature for 20 min. Staining for nuclei was performed using 100 μL per well of 2.5 μg/mL DAPI (Santa Cruz Biotechnology) for 5 min at room temperature then washed thrice with DPBS before staining with 100 μL of 0.165 μM Alexa Fluor 647 Phalloidin (Thermo Fisher) supplemented with 0.25% Triton X-100 for 20 min at room temperature. Cells were rinsed with DPBS and stored in 65% glycerol at 4 °C. Images were acquired using an EVOS FL microscope and analyzed using ImageJ.

**Double infection of HeLa cells and PNA treatment**. For the double infection of HeLa cells with target STm (SL1344-mCherry) and delivery STm (SL1344-holin) (Fig. 4d-f), single colonies were picked on solid media and grown overnight in 1 mL of LB broth supplemented with 30 μg/mL streptomycin and 100 μg/mL ampicillin. Overnight cultures were diluted 1:10 and regrown for 3 h. After regrowth the cultures were washed thrice with PBS and diluted to a concentration of $9 \times 10^6$ CFU/mL (MOI of 10). The SL1344-holin infection stock was split into no treatment and treatment; 10 μM of recA was added to the treatment condition and an equal volume of PBS was added to the no treatment. Both conditions were incubated for 45 min at 37 °C mimicking the conditions used in the single infection experiment for the PNA to enter the *Salmonella*. After incubation all infection stocks (target STm, delivery STm with no treatment, and delivery STm with treatment) were washed thrice with DPBS to remove any extracellular PNA. Each delivery-STm and target STm stock was combined at a 4:1 ratio to create the treatment and no treatment infection solutions and 50 μL was to infect the HeLa.

To ensure that the incubation ratio remained the same and PNA waste was minimized the DPBS washes were done with volumes under 1 mL resulting in variability of the actual MOI infected of each infection stock. To account for this variability each infection stock was serially diluted, plated, grown overnight, and CFUs counted to determine the exact number of bacteria that was used to infect for each condition. This extracellular CFU causing infection was measured for both delivery STm and target STm using florescence and is referred to as $CFU_{Delivery\ STm}^{Infected\ With}$ and $CFU_{Target\ STm}^{Infected\ With}$ respectively. For CFU counting the separate bacterial populations were separated by counting the Target STm as those colonies that fluoresced when excited by light at 587 nm in a light box and viewed through a 610 nm emission filter, and the delivery STm as colonies showing lack of fluorescence at those wavelengths. At 18 h post infection wells for CFU analysis were washed thrice with 300 μL PBS and lysed with 30 μL of 0.1% Triton X-100 for 15 min at room temperature. After 15 min, 270 μL of PBS was added to each well then serially diluted and plated on sold LB media supplemented with 40 μg/mL streptomycin. Plates were incubated for 16 h at 37 °C and CFU per milliliter determined. The intracellular CFU post lysis was measured for both SL1344-holin and target STm using florescence and is referred to as $CFU_{Delivery\ STm}^{Lysed}$ and $CFU_{Target\ STm}^{Lysed}$ respectively. To account for variability of MOI we normalized the CFU of SL1344-holin and target STm cells using the ratio $\left(\frac{CFU_{Target\ STm}^{Lysed}}{CFU_{Target\ STm}^{Infected\ With}}\right)$ and $\left(\frac{CFU_{Delivery\ STm}^{Lysed}}{CFU_{Delivery\ STm}^{Infected\ With}}\right)$ respectively.

**Effect of PNA after 45 min of incubation during HeLa infection**. SL1344 were cultured as described above and split 1:10 and regrown for 3 h. Samples were rinsed with PBS and diluted to a concentration equivalent to an MOI of 10 for a 24 h grown culture of HeLa cells in 100 μL. PNA (100 μM) was added at 5 μL to 45 μL of bacteria and incubated for 45 min at culturing conditions. Samples (10 μL) were taken at $t = 0$ (before PNA addition) and $t = 45$ min (after PNA treatment), and serially diluted, plated, and grown for 16 h at 37 °C and CFU counted.

**Double infection of RAW264.7 cells and PNA treatment**. Infection and treatment of RAW264.7 cells, as seen in Fig. 4g, was done following the same protocol as the HeLa double infection, but with a few modifications. The infection itself was done using ESBL KPN as the target bacteria and SL144-holin as the delivery STm, with ESBL KPN grown in CAMHB. After regrowth ESBL KPN was diluted to $1 \times 10^6$ CFU/mL (1 MOI in 50 µL) in DMEM + 10 %FBS and delivery STm was diluted to $1 \times 10^7$ CFU/mL (10 MOI in 50 µL) in DPBS. Delivery STm was split into treatment and no treatment, incubated with PNA (α-rpsD), delivery STm rinsed, and RAW264.7 cells rinsed as described previously. 50 µL of each diluted bacteria was added to RAW264.7 cells and the infection was synchronized by centrifugation at 200xg for 5 min prior to 45 min incubation. Gentamicin protection, incubation, and lysis was all done as described previously. CFU colonies were plated on LB containing 50 µg/mL Kanamycin, which ESBL KPN is resistant to but delivery STm is not, and LB containing 100 µg/mL ampicillin, which both are resistant to.

**Statistics and reproducibility**. Error bars represent one standard deviation of biological replicates. In all cases, significance designated with an asterisk (*) is defined as $p < 0.05$ for a 95% confidence interval. Significance of S values was determined using a one-sample $t$ test with a hypothesized mean of zero while all other statistical tests were done using a two-sided Student's $t$ test. Determination of Pearson's coefficients and their $p$ values, in Figs. 3a and 3q, were done using the statistical software JMP Pro 15; all other statistical tests were done in Excel.

**Reporting Summary**. Further information on research design is available in the Nature Research Reporting Summary linked to this article.

## Data availability
All data needed to evaluate the conclusions in the paper are present in the paper and/or the Supplementary Materials, including source data in Supplementary Data 1. Additional data available from authors upon request. Sequence data for the MDR clinical isolates used to identify antibiotic resistance genes and analyze PNA homology have been deposited in GenBank with the accession codes WWEV00000000.1, MSDR00000000.1, WWEX00000000.1, WWEY00000000.1, and WWEW00000000.1 for MDR E. coli, CRE E. coli, ESBL KPN, NDM-1 KPN, and MDR STm respectively.

## Code availability
The PNA Finder toolbox[76] is available at https://github.com/taunins/pna_finder and requires Python 3.7, Bowtie 2 (version 2.3.5.1)[28], SAMtools (1.9)[29], and BEDTools (v2.25.0)[30]. To run on a Windows operating system a Window-compatible bash shell is required, and Mac operating systems are not currently supported.

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

## Acknowledgements
We acknowledge financial support from W. M. Keck Foundation and DARPA Young Faculty Award (D17AP00024) and National Aeronautics and Space Administration (NASA) Cooperative Agreement Notice––Translational Research Institute (TRISH) award number NNX16A069A to A.C., NSF Graduate fellowship (DGE 1144083) to C.M. C. We thank RY Young from Texas A&M University for gifting the pRG1 plasmid. We thank Ryan Felton and Teruna J. Siahaan from The University of Kansas for the synthesis of PNA for toxicity studies in HeLa (Fig. S17). We thank Dr. Thomas Lee, a director of the Central Analytical Laboratory at the University of Colorado, Boulder for LC–MS analysis of the samples. The purchase of the Synapt G2 HD mass spectrometer was made possible with NIH grant S10-RR026641. The views, opinions, and findings contained in this article are those of the author and/or should not be interpreted as representing the official views or policies, either expressed or implied, of the Defense Advanced Research Projects Agency or the Department of Defense. All data needed to evaluate the conclusions in the paper are present in the paper and/or the Supplementary Materials. Additional data available from authors upon request.

## Author contributions
K.A.E., C.M.C., T.R.A., and J.K.C, conducted all experiments. N.E.M. provided the clinical isolates. K.E.E and T.R.A. performed the de novo assembly and sequence analysis for the clinical isolates. T.R.A. performed PNA synthesis and bioinformatics analysis and PNA off-target search across bacterial genomes. P.B.O. performed the protein interaction network analysis. A.C., C.M.C., K.A.E., and J.K.C., analyzed the experimental data. K.A.E., J.K.C., C. M.C., and A.C. analyzed HeLa infection data. A.C., C.M.C., K.A.E., P.B.O., and T.R.A. wrote the paper. All the authors discussed the results and edited the manuscript.

## Competing interests
The authors (A.C., and C.M.C.), (A.C., K.A.E., and J.K.C.), and (A.C. and T.A.) have filed patents on this technology. A.C. is the founder of a start-up company Sachi Bioworks, Inc. based on this technology.
