## [Peer Review File · Communications Biology]

Reviewers' comments:

Reviewer #1 (Remarks to the Author):

In this manuscript, the authors reported a "FAST" platform of antisense therapeutic strategy that targets MDR bacterial pathogens with the use of peptide nucleic acids (PNAs). Firstly, sequence-specific PNAs were bioinformatically designed and then PNAs' of those without off-targets were chemically synthesized with a terminal cell-penetrating peptide. Then the PNAs were tested for their ability to inhibit bacterial growth. The PNAs α -rpsD and α -lexA were found to be the most potent candidates against the pathogens tested. Surprisingly, there were no correlations between antibiotic potency and each gene target's corresponding mRNA abundance level and/or protein secondary structure. But there seems to be a correlation between the potency and protein network interactions mapped by STRING network analysis. Then the PNAs that showed minimal growth inhibition were tested for synergistic activity in combination with conventional antibiotics. Tetracycline and gentamicin were found to be synergistic with some of the PNAs. Lastly, the impeded mammalian cellular uptake of potent PNAs has been improved using Type III secretion system (T3SS) of Salmonella; a first probiotic prototype delivery system.

With the world at massive risk due to the reducing utility of our current antibiotics, the need for new strategies is paramount. This manuscript presents an interesting platform to overcome widespread bacterial resistance. However, there are some major concerns that the authors should consider and it is not worthy for publication in its present form.

Major inconsistencies:

1. The authors should provide the actual chemical characterization data of the synthesized PNAs. This would really help the scientific community to replicate the data.
2. Antimicrobial studies – The antimicrobial studies have been performed in an unconventional manner. It would be much better if they present the actual MIC₉₀ values for PNAs and this way it would be appropriate to compare the potencies.
3. Combination studies – Authors should consider performing in a checkerboard manner; the most conventional way to assess the potency of antibiotic combinations.
(<https://www.sciencedirect.com/science/article/abs/pii/S0966842X16300725>;
<https://www.nature.com/articles/s41579-018-0141-x>)
4. Though authors have seen some correlation between PNAs' antibiotic potency and protein-network interactions there was no direct correlation with corresponding mRNA abundance level and/or secondary structure which was expected. I was wondering if any auxiliary mechanism that imparts for antibiotic activity.
5. How the PNAs are assisting the conventional antibiotics in combination studies?? What is the mechanism??
6. PNA α -gyrB potentiated both chloramphenicol (CHL) and tetracycline (Tetra). As PNA α -gyrB directs the functional target of fluoroquinolones (FLQ); Would CHL and Tetra be potentiated by FLQ??

7. Intracellular infection studies - Generally macrophages are used for intracellular activity studies. HeLa cell line is not appropriate for infection studies. The authors should consider testing the whole experiment with macrophages.

8. What happens if you append a long chain to PNA?? In intracellular activity studies?? Long lipophile might assist for intracellular uptake and in this case, the probiotic prototype may not be needed.

Reviewer #2 (Remarks to the Author):

Authors present a potential of the FAST platform in treating MDR and a T3SS delivery approach to treat intracellular infections of bacteria. It is a nice work and good data. However, I have a few points I would like to raise with the authors. The paper could be published with minor revised.

Major issues:

1. An obvious question in intracellular test is why use Salmonella as delivery (Delivery STm) to treat Salmonella (Target STm)(Fig. 4D). A non-T3SS bacteria (E. coli MG1655) was used as the Delivery strain to treat Target STm, and the result shows no reduction in Target STm(Fig. 4F); To prove Delivery STm system more efficiently, it is useful to add another control that Delivery STm treat non-STm intracellular bacteria(for instance: Staphylococcus aureus)

2. Page9, Line11, Authors directly mixing PNA, STM and HeLa cells for 45min. I am curious why you do not load PNA to Delivery Stm in advance, and then incubate with HeLa cells for 45min. It is convenient to calculate the PNA loading rate of delivery Stm. Indeed, The manuscript would also benefit from calculating the loading rate.

3. Page 7, Line 17, "the more interconnected a network is the more susceptible it is to PNA treatment"

Page 8, Line 6, "targeting genes that have less interconnected protein interaction networks could give rise to synergy with antibiotics"

According to the above two sentences, my understanding is that the more interconnected networks the gene is, the better antibacterial activity the α -gene is. but when PNA is combined with antibiotics, the less interconnected networks the gene is, the more synergy with antibiotics the α -gene is. It seems a bit contradictory, Explain more if possible.

4. The FAST platform: 71 essential genes and 243 non-essential genes were screened out (Fig. 1A). Though their activity is different : α -ffh(4/5), α -lexA(4/5), α -acrA(3/5), All three chosen genes (α -ffh, α -lexA, α -acrA) just work(Figure S5). However, it needs to find another target genes from 314 genes and synthesize the corresponding PNAs if none of three PNAs is effective(α -ffh(0/5), α -lexA(0/5), α -acrA(0/5)). Further, Can I think that the main function of the FAST platform is to narrow the scope of the target genes? If not, how can it ensure that the selected target gene is definitely effective?

Minor issues:

1. Why the ratio is 4 (Delivery STm): 1(Target STm)(Page 18, Line 13)?

2. Figure S3A, the meaning of green curve should be explained.

3. Page 30, Line 4: "Supplementary Figure S12" should be "Supplementary Figure S7".

4. In "PNA growth assay", untreated group grow to 16h (Fig. 2E-F), but in "Potentiation of antibiotics

with PNAs”, untreated group grow to 24h (Fig. 3B-M), why ? In “Figure S8” “at 16 hours as compared to no treatment” Please confirm.

Reviewer #1 (Remarks to the Author):

In this manuscript, the authors reported a “FAST” platform of antisense therapeutic strategy that targets MDR bacterial pathogens with the use of peptide nucleic acids (PNAs). Firstly, sequence-specific PNAs were bioinformatically designed and then PNAs’ of those without off-targets were chemically synthesized with a terminal cell-penetrating peptide. Then the PNAs were tested for their ability to inhibit bacterial growth. The PNAs α -rpsD and α -lexA were found to be the most potent candidates against the pathogens tested. Surprisingly, there were no correlations between antibiotic potency and each gene target’s corresponding mRNA abundance level and/or protein secondary structure. But there seems to be a correlation between the potency and protein network interactions mapped by STRING network analysis. Then the PNAs that showed minimal growth inhibition were tested for synergistic activity in combination with conventional antibiotics. Tetracycline and gentamicin were found to be synergistic with some of the PNAs. Lastly, the impeded mammalian cellular uptake of potent PNAs has been improved using Type III secretion system (T3SS) of Salmonella; a first probiotic prototype delivery system.

With the world at massive risk due to the reducing utility of our current antibiotics, the need for new strategies is paramount. This manuscript presents an interesting platform to overcome widespread bacterial resistance. However, there are some major concerns that the authors should consider and it is not worthy for publication in its present form.

We thank the reviewer for their comments and have updated the manuscript to address their concerns as discussed below.

Major concerns:

1. The authors should provide the actual chemical characterization data of the synthesized PNAs. This would really help the scientific community to replicate the data.

We thank the reviewer for their suggestion and have included the chemical characterization for all PNA in supplement Figures S2 and S3.

2. Antimicrobial studies—The antimicrobial studies have been performed in an unconventional manner. It would be much better if they present the actual MIC90 values for PNAs and this way it would be appropriate to compare the potencies.

We thank the reviewer for their suggestion and have evaluated each PNA for their 90% growth inhibition concentration for the clinical isolates and have included these values in a table in Figure 2G as well as the growth curves in Figure S8.

*3. Combination studies—Authors should consider performing in a checkerboard manner; the most conventional way to assess the potency of antibiotic combinations.
(<https://www.sciencedirect.com/science/article/abs/pii/S0966842X16300725>; <https://www.nature.com/articles/s41579-018-0141-x>)*

We thank the reviewer for their suggestion and have incorporated a checkerboard analysis of synergy for each of the PNA and antibiotic combinations done. These results can be seen in Figure 3N-P, supplemental figures S12-15, and discussed on **page 8 lines 8 to 11**. Our results show that there seems to be a critical point at higher PNA and antibiotic concentrations and almost no synergy at lower concentrations. The combination of tetracycline and α -acrA in ESBL KPN had to be done at lower tetracycline concentrations than done previously because during the course of testing these combinations the clinical isolate seems to have adapted and decreased its resistance to tetracycline.

4. Though authors have seen some correlation between PNAs' antibiotic potency and protein-network interactions there was no direct correlation with corresponding mRNA abundance level and/or secondary structure which was expected. I was wondering if any auxiliary mechanism that imparts for antibiotic activity.

We thank the reviewer for their comment and agree that this is a topic we intend to explore further. There are numerous mechanisms that could impart different levels of efficacy to each PNA antibiotic the most obvious being the target gene's mechanism which is why we looked at a range of targets including unconventional antibiotic targets and essential and non-essential genes. We further attempted to explore other parameters that might relate to the efficacy and found no correlation with mRNA abundance but did find a correlation with the target's protein network interactions. These are three parameters that we have looked at so far when testing our new FAST platform. The adaptive design of the PNA Finder toolbox allows for further exploration into what parameters are significant for efficacy of our PNA antibiotics and we intend to continue to add more parameters to test and evaluate, beyond the parameters in this manuscript (gene target, mRNA abundance level, and protein network interactions), in future work. We have updated the text on **pages 11 line 31 to page 12 line 3** to include a statement describing how continued testing using the FAST platform will explore the auxiliary mechanisms that impart antibiotic activity to the PNAs and improve the PNA Finder toolbox.

5. How the PNAs are assisting the conventional antibiotics in combination studies?? What is the mechanism??

We thank the reviewer for their comment and have updated the text to further discuss this on **page 8 line 27 to 30**. While Dryselius et al. and Castillo et al. have shown that the combination of PNA and antibiotics that target similar pathways show greater synergistic effects compared to those targeting unrelated pathways, we have previously shown, across numerous genetic perturbations in combination with antibiotics, that by targeting both related and unrelated pathways one can achieve synergistic effects (Courtney et al. (2017) Science Advances, Otoupal et al. (2018) Nature Communications Biology (**ref no. 48**), and Aunins et al. (2020) PNAS (**ref no. 31**)).

The updated text includes two new citations illustrating these results. The first new citation by Otoupal et al. (2018) (**ref no. 48**) illustrates how numerous fitness-neutral perturbations targeting different pathways impart negative epistasis when multiplexed. This phenomenon was first tested using CRISPR Cas-9 gene expression perturbations and was then tested using

PNA. Combining PNAs targeting different pathways that show little or no toxicity on their own or in combination, show a relative reduction in fitness when in combination with an antibiotic, chloramphenicol, at an antibiotic concentration that has no toxicity on its own or in combination with each PNA individually. The relevant figure from this source within this passage is Fig. 6 and is shown below.

Fig. 6 CHAOS increases the antibiotic susceptibility of clinically isolated CRE *E. coli*. A CRE isolate of *E. coli* exhibiting resistance to at least 11 antibiotics above CLSI breakpoint levels was isolated from a clinical infection (Supplementary Table 4). We focused on applying CHAOS induced epistasis to resensitize this isolate to chloramphenicol. a A new set of four universally conserved bacterial genes were perturbed using PNA to demonstrate applicability outside of CRISPR interference and towards clinically relevant infections. PNA structure consists of a peptide backbone connecting nucleosides analogous to DNA and linked to a cell penetrating peptide. These molecules are able to enter bacteria and anneal tightly to analogous mRNA sequences, allowing for targeted blockage of protein translation. Chloramphenicol-resistant CRE *E. coli* was exposed to 2.5 µM of four unique PNAs either individually or in combination (for a total concentration of 10 µM PNA) for 24 h, after which cells were plated on both plain LB agar, as well as clinically-relevant levels of chloramphenicol to determine viable cells. b CFU analysis of CRE *E. coli* after exposure to PNA treatment demonstrates CHAOS's effectiveness. Exposure to PNA ffh resulted in a ~16-fold reduction in viable cells with respect to no PNA treatment, while the remaining PNAs exhibited largely no effect under both conditions. Combination of all 4 PNAs exacerbated chloramphenicol's toxicity and gave rise to ~110-fold reduction in viable cells with respect to no PNA treatment in an apparently epistatic fashion even at sub-resistance levels. *P* values were calculated using two-tailed type II *t*-test

The second new citation by Otoupal et al. (2019) (ref no. 49) further explored potentiating antibiotics using CRISPR-dCas9 gene perturbations and/or PNA. Synergy was seen in target pathways that were similar and in target pathways that were different using both CRISPR-dCas9 gene perturbations and PNA. These articles highlight that multiple perturbations to bacteria's homeostatic functions can create a synergistic effect on their fitness that is due to the combination of multiple fitness burdens exacerbating the bacteria's defenses rather than the similarities in the targeted pathways.

6. PNA *α*-*gyrB* potentiated both chloramphenicol (CHL) and tetracycline (Tetra). As PNA *α*-*gyrB* directs the functional target of fluoroquinolones (FLQ); Would CHL and Tetra be potentiated by FLQ??

We thank the reviewer for their suggestion and have explored this by doing a checkerboard analysis of CHL and ciprofloxacin synergy on ESBL KPN and have included the results in Figure S16 and discussed this in the main text on page 8 lines 11-16. We saw no significant synergy in any of the combinations we explored.

7. Intracellular infection studies - Generally macrophages are used for intracellular activity

studies. HeLa cell line is not appropriate for infection studies. The authors should consider testing the whole experiment with macrophages.

We thank the reviewer for their suggestion and have evaluated the effectiveness of treating an intracellular infection with our Deliver STm carrying PNA in RAW264.7 cells and show effective treatment of a *Klebsiella pneumoniae* infection utilizing our method. Figure 4 panel G has been updated to include these results as well as the methods and results sections.

8. What happens if you append a long chain to PNA?? In intracellular activity studies?? Long lipophile might assist for intracellular uptake and in this case, the probiotic prototype may not be needed.

We thank the reviewer for their comment and understand that this, along with CPPs, nanoparticles, and cationic lipids, is another option for increasing intracellular uptake of PNA that is currently being explored in the scientific community (Gupta et al. (2016) Journal of Controlled Release, Hassane et al. (2009) Cellular and Molecular Life Sciences, Shiraishi et al. (2008) Nucleic Acids Research, and Swenson (2020) Chemical Communications). While this is an option, we believe our work highlights a novel alternative, a T3SS mediated bacterial delivery system, further expanding the repertoire of PNA delivery options and that pursuing a long lipophilic chain as a delivery method is beyond the scope of this paper.

Response to the Second Referee:

Reviewer #2 (Remarks to the Author):

Authors present a potential of the FAST platform in treating MDR and a T3SS delivery approach to treat intracellular infections of bacteria. It is a nice work and good data. However, I have a few points I would like to raise with the authors. The paper could be published with minor revised.

We thank the reviewer for their constructive feedback and comments, which we have used to improve the manuscript. We provide our responses to the comments and the changes made below.

Major concerns:

1. An obvious question in intracellular test is why use Salmonella as delivery (Delivery STm) to treat Salmonella (Target STm) (Fig. 4D). A non-T3SS bacteria (E. coli MG1655) was used as the Delivery strain to treat Target STm, and the result shows no reduction in Target STm (Fig. 4F); To prove Delivery STm system more efficiently, it is useful to add another control that Delivery STm treat non- STm intracellular bacteria(for instance: Staphylococcus aureus)

We thank the reviewer for their comment and based on their suggestions have evaluated the efficacy of Delivery STm to treat a non-STm intracellular infection: ESBL KPN in macrophage RAW264.7 cells. Figure 4 panel G and the main text, page 11 lines 8-19, have been updated to include these results showing a significant reduction in intracellular load of ESBL KPN.

2. Page9, Line11, Authors directly mixing PNA, STM and HeLa cells for 45min. I am curious why you do not load PNA to Delivery Stm in advance, and then incubate with HeLa cells for 45min. It is convenient to calculate the PNA loading rate of delivery Stm. Indeed, The manuscript would also benefit from calculating the loading rate.

We thank the reviewer for their feedback and agree that it would be better practice to load the Delivery STm with PNA prior to infecting HeLa cells. In the first intracellular infection experiment validating the ability of Salmonella to carry PNA into the host cell we did not preload the bacteria with PNA. This was initially to reduce any variation in the MOI of infection from having to repeatedly rinse and pellet the bacteria after incubating with PNA to remove any extracellular PNA prior to infecting into HeLa cells, but as we transitioned to our second intracellular infection experiment we revised our protocol to preload the PNA into the bacteria as we discovered that the rinsing and pelleting steps did not drastically change the MOI.

In addition, we have updated the main text on page 9 line 30 to page 10 line 2 to include resources that have shown that at 45-minutes PNA uptake is observed (ref. 51, 55, 56). Where Goltermann et al. (2019) show that (KFF)₃K-PNA conjugate uptake is limited by the bacterial inner membrane and Eriksson et al. (2002) show that inner cell membrane permeabilization begins to saturate at 10 minutes by (KFF)₃K-PNA conjugates. These results in combination with results by Nikravesh et al. (2007) detailing that cell-associated (KFF)₃K-

PNA conjugate concentration reaches extracellular media concentrations by 30 minutes and is retained for up to 5 hours indicates that we have sufficient intracellular concentration of PNA in our Delivery STm, likely equaling the initial loading concentration.

3. Page 7, Line 17, “the more interconnected a network is the more susceptible it is to PNA treatment”

Page 8, Line 6, “targeting genes that have less interconnected protein interaction networks could give rise to synergy with antibiotics”

According to the above two sentences, my understanding is that the more interconnected networks the gene is, the better antibacterial activity the α -gene is. but when PNA is combined with antibiotics, the less interconnected networks the gene is, the more synergy with antibiotics the α -gene is. It seems a bit contradictory, Explain more if possible.

We thank the reviewer for their comment and have updated the text to clarify and further discuss this on **page 8 lines 20-23**. The phenomenon elucidated by our results suggests to us that as a monotherapy it is important to have far reaching effects meaning that when one protein is targeted numerous other proteins are affected as well. Whereas when using a combination, we are looking for a synergistic effect of two treatments. If monotherapy is already strong then the likelihood of observing synergy when combined with another target is low. We generally observe that combining targets that individually have a lower number of connections collectively have a stronger impact when combined. This is likely due to the fact that when targeted together it leads to perturbation of larger network that extends the range of effects. In contrast, targeting a protein with more network connections naturally has a stronger therapeutic effect by itself, which is not improved upon by synergy.

4. The FAST platform: 71 essential genes and 243 non-essential genes were screened out (Fig. 1A). Though their activity is different: α -ffh(4/5), α -lexA(4/5), α -acrA(3/5), All three chosen genes (α -ffh, α -lexA, α -acrA) just work (Figure S5). However, it needs to find another target genes from 314 genes and synthesize the corresponding PNAs if none of three PNAs is effective (α -ffh(0/5), α -lexA(0/5), α -acrA(0/5)). Further, Can I think that the main function of the FAST platform is to narrow the scope of the target genes? If not, how can it ensure that the selected target gene is definitely effective?

We thank the reviewer for their feedback and agree that improving the PNA Finder toolbox within the FAST platform to suggest PNAs that we expect to be more effective is a topic we intend to explore further. We have updated the text on **page 11 lines 8 to 19** to include a statement describing how continued testing using the FAST platform will explore the mechanisms that impart antibiotic activity to the PNAs and improve the PNA Finder toolbox. Currently the scope of the PNA Finder toolbox is to provide a list of targets that we expect to be effective based on predictive homology and be species specific. We believe that to ensure that the selected target genes are effective requires the incorporation of more parameters related to the PNA efficacy which is why we explored three of those parameters in this initial test of the FAST platform: gene target mechanism (tested nontraditional antibiotic targets and essential and nonessential genes), mRNA abundance levels, and protein network interactions. The FAST platform is designed to be adaptive and as such as more PNAs are tested and correlations (gene target mechanism, mRNA abundance level, protein network interaction,

etc) are determined the PNA Finder toolbox will be updated to provide a list of PNAs based on predictive efficacy.

Minor concerns:

1. Why the ratio is 4 (Delivery STm): 1(Target STm) (Page 18, Line 13)?

We believed that providing more therapeutic treatment, analogous to a therapeutic dose dependent response, would be more efficacious and as such chose an arbitrary ratio of 4:1 for initial studies and found that it was effective and maintained that ratio for consistency.

2. Figure S3A, the meaning of green curve should be explained.

This has been fixed and removed.

3. Page 30, Line 4: "Supplementary Figure S12" should be "Supplementary Figure S7".

This has been fixed.

4. In "PNA growth assay", untreated group grow to 16h (Fig. 2E-F), but in "Potentiation of antibiotics with PNAs", untreated group grow to 24h (Fig. 3B-M), why? In "Figure S8" "at 16 hours as compared to no treatment" Please confirm.

Due to the fact that with combination treatment it is possible that the combined effects of two treatments do not inhibit bacteria growth but instead significantly slow bacterial growth we chose to evaluate combination at a longer time point of 24 hours. Synergy S values were evaluated at 24 hours correlated with the antibiotic combination data, bar graphs, that are at 24 hours. The text in the caption of Figure 3 has been updated to clarify this, **page 32 lines 15 to 16.**

REVIEWERS' COMMENTS:

Reviewer #1 (Remarks to the Author):

This manuscript demonstrates the utility of FAST Platform to counter MDR pathogens. The authors have responded adequately to the original critics however I still have a few concerns. In view of the great general interest in the field of novel antibiotic concepts against resistant bacteria; overall, this thoroughly revised manuscript presents a novel concept that is worth publishing after addressing the mentioned comments.

1. Fig. 2C – Authors are recommended to present the actual MICs of the listed MDR pathogens (maybe as supplementary table), and this data is necessary to correlate the synergistic value (S) of the combination of PNA with the conventional antibiotics against the listed MDR pathogens.
2. Most of the PNAs selected and synthesized for example α -gyrB, rpsD, recA, lexA, etc. can also be used to target Gram-positives as the corresponding genes are essential for their growth and these genes are functionally/structurally homologous to Gram-negatives. I wonder how these PNAs affect Gram-positive bacterial growth.
3. Authors should comment on the antagonism observed with meropenem and α -rpsD (Fig. S13B).
4. In response to my original comment 6: Authors have not performed the combination of ciprofloxacin (CIP) with tetracycline. Only performed Cip with chloramphenicol.

Fig. S16- Why authors used such low concentrations of Cipro in checkerboard assay? I understand that an MIC of $>1 \mu\text{g}/\text{mL}$ can be classified as ciprofloxacin-resistance, however, to know in general whether or not the combination is synergistic, a few high concentrations of the antibiotic are recommended. What is the actual MIC of Cipro against this particular pathogen?

5. Authors are advised to thoroughly check their text for any typos. Bacterial nomenclature should be maintained in italics.

Reviewer #3 (Remarks to the Author):

Basically no more new comments from me and my questions put out in first turn of reviewing were well addressed by authors one by one in this new revision with a full length of over 70 pages and another supplement materials of over 40 pages. Obviously it reached and met the critical standards for its acceptance and publication in this journal. I think so.

Reviewer #1 (Remarks to the Author):

This manuscript demonstrates the utility of FAST Platform to counter MDR pathogens. The authors have responded adequately to the original critics however I still have a few concerns. In view of the great general interest in the field of novel antibiotic concepts against resistant bacteria; overall, this thoroughly revised manuscript presents a novel concept that is worth publishing after addressing the mentioned comments.

We thank the reviewer for their comments and have updated the manuscript to address their concerns as discussed below.

Major concerns:

1. Fig. 2C – Authors are recommended to present the actual MICs of the listed MDR pathogens (maybe as supplementary table), and this data is necessary to correlate the synergistic value (S) of the combination of PNA with the conventional antibiotics against the listed MDR pathogens.

We thank the reviewer for their suggestion and have included a supplemental table (Table S2) that indicates the corresponding antibiotic MIC90s for all clinical isolates.

2. Most of the PNAs selected and synthesized for example α -gyrB, rpsD, recA, lexA, etc. can also be used to target Gram-positives as the corresponding genes are essential for their growth and these genes are functionally/structurally homologous to Gram-negatives. I wonder how these PNAs affect Gram-positive bacterial growth.

We thank the reviewer for their insight and agree that future work with Gram-positive bacteria is something we plan to explore but is outside the scope of this paper. Here we focus on treating five MDR clinical isolate Enterobacteriaceae. Additionally, peptide nucleic acids have been tested on Gram-positive bacteria and shown to inhibit growth inhibition in a number of publications including Dryselius et al. (2005), Nekhotiaeva et al. (2004), Liang et al. (2015), and Bai et al (2012).

3. Authors should comment on the antagonism observed with meropenem and α -rpsD (Fig. S13B).

We thank the reviewer for their suggestion and have updated the main text to note these results on **page 8 lines 20 to 21**. We did not speculate on the result as it is not a consistent trend at higher concentrations.

4. In response to my original comment 6: Authors have not performed the combination of ciprofloxacin (CIP) with tetracycline. Only performed Cip with chloramphenicol.

We thank the reviewer for their suggestion and insight into investigating the interaction between fluoroquinolones and chloramphenicol and/or tetracycline in response to the synergistic reaction seen between α -gyrB and CHL and TET. Following their suggestion, we investigated the interaction between CIP and CHL in the clinical isolate ESBL KPN and found no interaction.

We acknowledge that we did not also probe the interaction between CIP and TET and have updated the manuscript text to further discuss this on **page 8 lines 27 to 28**. We believe that further exploration of the mechanistic interactions behind the combination therapies is an avenue for further research but is beyond the scope of this paper. We believe the comparison between CHL and CIP and α -gyrb and CHL is a sufficient preliminary exploration in our work establishing and validating a new therapeutic platform.

Fig. S16- Why authors used such low concentrations of Cipro in checkerboard assay? I understand that an MIC of $>1 \mu\text{g/mL}$ can be classified as ciprofloxacin-resistance, however, to know in general whether or not the combination is synergistic, a few high concentrations of the antibiotic are recommended. What is the actual MIC of Cipro against this particular pathogen?

We thank the reviewer for their question. We chose to do CIP concentrations of 0, 0.5, and 1 $\mu\text{g/mL}$ in combination with CHL (0, 4, and 8 $\mu\text{g/mL}$) to be consistent with our comparison to our PNA and antibiotic synergy results. In our combination data we specifically explored low antibiotic concentrations, at or below the sensitive CLSI breakpoint, in combination with PNA that were not effective monotherapies to show re-sensitization of resistant bacteria to antibiotics when combined with PNA. In our combination data CHL was used at concentrations of 2, 4, and 8 $\mu\text{g/mL}$ and CIP was used at concentrations of 0.5, 1, and 2 $\mu\text{g/mL}$, therefore we explored combinations within those same concentration windows. The MIC of CIP for the clinical isolate used, ESBL KPN, is greater than 4 $\mu\text{g/mL}$.

5. Authors are advised to thoroughly check their text for any typos. Bacterial nomenclature should be maintained in italics.

We thank the reviewer for their suggestion and have reviewed and corrected any typos.

University of Colorado **Boulder**
DEPARTMENT OF CHEMICAL & BIOLOGICAL ENGINEERING

JSC Biotechnology Center, Campus Box 596
Boulder, CO 80309-0596
Phone: 303-735-6586
Fax: 303-492-8425
Email: chatterjee@colorado.edu

Response to the Second Referee:

Reviewer #2 (Remarks to the Author):

Basically no more new comments from me and my questions put out in first turn of reviewing were well addressed by authors one by one in this new revision with a full length of over 70 pages and another supplement materials of over 40 pages. Obviously it reached and met the critical standards for its acceptance and publication in this journal. I think so.

We thank the reviewer for their feedback and careful review which we used to improve the manuscript.